# *Plasmodium falciparum surf$_{4.1}$* in clinical isolates: From genetic variation and variant diversity to *in silico* design immunopeptides for vaccine development

**Nitchakarn Noranate**[1]*, **Jariya Sripanomphong**[1], **Fingani Annie Mphande- Nyasulu**[1], **Suwanna Chaorattanakawee**[2]

**1** Faculty of medicine, King's Mongkut Institute of Technology (KMITL), Ladkrabang, Bangkok, Thailand, **2** Department of Parasitology and Entomology, Faculty of Public Health, Mahidol University, Bangkok, Thailand

* noranate@gmail.com

**Data Availability Statement:** Data is provided in supplementary file S3.

## Abstract

SURFINs protein family expressed on surface of both infected red blood cell and merozoite surface making them as interesting vaccine candidate for erythrocytic stage of malaria infection. In this study, we analyze genetic variation of *Pfsurf$_{4.1}$* gene, copy number variation, and frequency of SURFIN$_{4.1}$ variants of *P. falciparum* in clinical isolates. In addition, secondary structure prediction and immunoinformatic were employed to identify immunogenic epitopes in humoral response as proposed vaccine candidates. Overall, our data demonstrate extensive polymorphism of SURFIN$_{4.1}$ in both genetic and protein level. The *surf$_{4.1}$* gene showed extensive genetic variation with total of 447 polymorphic sites with maximum of three variants as well as singlet/triplet bases indels and mini/microsatellites in the coding sequence. The exon1 encoding extracellular region exhibited higher variation compared to exon2 which coding for intracellular domain. Interestingly, selective pressure was detected on both extracellular region (Var1 and Var2) as well as intracellular region (WRD2 and WRD3). Importantly, extensive full gene analysis suggests adenosine insertion at three key points nucleotide bases (nt 2409/2410, 3809/3810, and 4439/4440) of exon2 could lead to frameshift mutation resulted in four different SURFIN$_{4.1}$ variants (TMs, WD1, WD2 and WD3). The SURFIN$_{4.1}$ variant TMs was the most observed type with 67% frequency (51/76). Along with more than one copy number of *surf$_{4.1}$* gene was observed with frequency of 13% (9/70). Despite substantial polymorphism, analysis of relatedness within *P. falciparum* population using full coding sequence was able to group SURFIN$_{4.1}$ protein into five distinct clades and reduced into four clades when using only exon1 coding sequence. Also, predicted secondary structure revealed conserved structure of five helix domains of extracellular region which similar among four SURFIN$_{4.1}$ variant types. In addition, *in silico* design eight immunopeptides derived from SURFIN$_{4.1}$, four of which are highly conserved and four of dimorphic epitopes, as potential vaccine candidates.

**Funding:** This work is supported by King Mongkut's Institute of Technology Ladkrabang [Grant number 2563-0216003]. The funding source had no role in study design, analysis or interpretation of data, preparation of the manuscript or the decision to publish.

**Competing interests:** The authors declare that they have no competing interests.

## Introduction

The erythrocytic stage of malaria causes symptoms in infected individuals. During, *Plasmodium falciparum (P. falciparum)*, a malaria parasite, development inside red blood cells, several parasite-produced proteins are exposed on the RBC surface i.e. the surface-associated interspersed proteins (SURFINs), repetitive interspersed families of polypeptides (RIFIN), subtelomeric variant open reading frame (STEVOR), and erythrocyte membrane protein 1 (EMP1) [1, 2]. Erythrocyte surface exposure allows these proteins to mediate interaction between infected red blood cell (iRBC) with other cells, including uninfected RBCs and endothelial cells, and to serve as targets for immune responses [3]. After successful development, merozoites were released and re-invade new RBC continuing rounds of erythrocytic development. Parasitized red blood cells and newly released merozoites are two main key players in the erythrocytic stage of malaria infection and are important target for blood stage vaccine development.

The SURFIN proteins present on both infected erythrocytes and merozoites are encoded by a family of 10 surf genes located within or close to the sub-telomeres of five chromosomes. The SURFINs possess component of two major virulence factors, the N-terminal cysteine rich domain (CRD) similar to *P. vivax* VIR and the C-terminal tryptophan rich domains (WRDs) related to *P. falciparum* EMP1 and *P. knowlesi* schizont infected cell agglutination variant antigen (SICAvar) [4]. Naturally acquired antibody against the SURFINs were found in individual exposed to natural infection and recognized regions which located on both extracellular region and intracellular domain [5]. These make SURFINs to be ideal candidates for vaccine development for antimalaria erythrocytic stage.

However, SURFIN$_{4.1}$ is highly polymorphic, particularly at the variable region of the extracellular domain located just before transmembrane region [6, 7]. Also, the observation of SURFIN$_{4.1}$ different variants were reported [7, 8]. The *surf$_{4.1}$* gene located on chromosome 4 of *P. falciparum* is comprised of two exons separated by one intron [8]. High diversity of variable regions in exon 1 were reported in *P. falciparum* field isolates (Kenya and Thailand) and suggested *surf$_{4.1}$* gene is under immune selective pressure [6, 7, 9]. In addition, the extent of genetic variation and selection on exon 2 remains undefined. Moreover, copy number variation (CNV) of *surf$_{4.1}$* among *P. falciparum* strains had been reported [10]. As well as *surf$_{4.1}$* CNV was found different among populations which might be due to level of malaria transmission [7]. CNV had been reported in association with specific *P. falciparum* drug resistance phenotypes, erythrocyte invasion, cytoadherence and transcriptional regulation [11, 12]. These indicating CNV plays role in malaria parasite adaptation [13, 14]. Still the causal of CNV of *surf$_{4.1}$* has not been verified.

The objective of this study was to analyze genetic variation of *Pfsurf$_{4.1}$* gene, frequency of SURFIN$_{4.1}$ variants and copy number variation in Thai clinical isolates. In addition, immunoinformatic was employed to identify SURFIN$_{4.1}$ immunogenic epitopes as proposed vaccine candidates.

## Material and methods

### 1. Study samples

A total of 76 *P. falciparum*-infected blood samples collected between 1998 and 2000 were analyzed in this study. Samples were obtained from patients living in northwest Thailand near the Myanmar border. Patients underwent clinically appropriate treatment based on presenting clinical features at the hospital for Tropical Disease, Faculty of Tropical Medicine, Mahidol University, Bangkok, Thailand. *P. falciparum* infection was diagnosed by microscopic

examination of giemsa-stained thick and thin blood films. Patients were classified clinically as uncomplicated (n = 23) and complicated malaria (n = 53) according to the criteria of the World Health Organization [15]. Uncomplicated or mild malaria was characterized by a positive blood smear, fever without other identified cause of infection, and the absence of manifestations of severe malaria as described below. Complicated or severe malaria was characterized by one of the following symptoms: high parasitemia (>100,000 parasites/μl), hypoglycemia (glucose < 2.2 mmole/L), severe anemia (haematocrit < 20% or haemoglobin < 7.0 g/dl), increased serum creatinine levels (> 3.0 mg/dl) and unrousable coma. All subjects were 13 years old or older. Prior to enrollment, written informed consent was obtained from all participants or their parents or guardians for those under 18 years of age. This study was approved (exempted) by the institutional review boards of King's Mongkut Institute of Technology (EC-KMITL_63_054), Thailand.

## 2. *Plasmodium falciparum surf₄.₁* gene: Polymerase chain reaction (PCR) and NGS-based BTSeq™ sequencing

Primers for amplify full-length *surf₄.₁* were designed based on *P. falciparum* 3D7 sequence, (S41NTMF 5'-AAAGTTTTATTAAACCAGAAATGTAAAC-3' and S41R 5'- CTCCCATTCTGTA ATCTTGTTCCTCTTC-3') (PF3D7_0402200, Plasmodb). The approximate size of *Pfsurf₄.₁* was 6757 base pairs (bp). Nested PCR with the same set of forward (S41NTMF) and reverse primer (S41R) were performed with following conditions: initial denaturation step at 96°C for 5 minutes, followed by 35 amplification cycles of denaturation at 96°C for 15 seconds, annealing at 58°C and extension step at 68°C for 7 minutes and the final extension step at 68°C for 7 minutes and reaction were held at 4°C. PCR amplicons of full-length *surf₄.₁* gene were amplified using PrimeSTAR®GXL DNA polymerase (Takara Bio, Shiga, Japan). PCR products were purified using the PureLink™ PCR purification kit (Invitrogen by Thermo Scientific, Waltham, MA). The amplified PCR products were sequenced using BTSeq™ (Celemics, Seoul, Korea) which is NGS-based molecular barcoding technology allowing for the analysis of long-range PCR amplicons [16]. *P. falciparum 3D7* was used as reference in both PCR and sequencing steps.

## 3. Copy number of *Pfsurf₄.₁* gene by quantitative real-time PCR (qPCR)

Copy number variation of *Pfsurf₄.₁* gene was examined by the comparative $C_T$ method using *Pfama1* gene as a reference gene locus through a modification of previously described methods [7]. Briefly, real-time PCR was performed in CFX connect Real-Time PCR system (Bio-rad, Hercules, CA), using Luna Universal qPCR Master Mix (New England BioLabs, Ipswich, MA). Primers of *Pfsurf₄.₁* were PFD0100c.rtF2 (5'-TAAGAACAGAACATAATTATGATAA-3') and PFD0100.rtR1 (5'-CAATCCTGTTCTGCATATTTTATG-3') and of *Pfama1* (PF3D7_1133400) were fAMA1-RT.F1 (5'-AAGACGAAAATACATTACAACACGCA-3') and fAMA1-RT.R1 (5'-CTACTCTTATACCTGAACCATGAACT-3'). The thermocycler program for *Pfsurf4.1* and *Pfama1* genes were performed with following conditions: initial denaturation step at 95°C for 60 seconds, followed by 40 amplification cycles of denaturation at 95°C for 15 seconds and extension step at 60°C for 30 seconds. *P. falciparum* 3D7 known to harbor a single copy of both *surf₄.₁* and *ama1* genes [7, 10] were used as a calibrator (a control sample). Culture adapted *P. falciparum* 3D7 with 2% parasitemia were used to prepare genomic DNA using QIAamp DNA Blood mini kit following the manufacture's manual instruction (Hilden, Germany). DNA concentration was measured by Implean NanoPhotometer® N60/N50 (Munich, Germany). At the end of each reaction, the quality of the PCR amplified products was validated by the melting curve method. The efficiency of qPCR for both *surf₄.₁* and *ama1* were

0.99 and 0.87 as determined by serial 10-fold diluted *P. falciparum* 3D7 DNA (10 - $1\times10^{-5}$ ng). In all experiments performed, genomic DNA of *P. falciparum* 3D7 and non-template control (NTC) were included, and all samples were tested in triplicate. Calculate quantification of the copy number change of target gene relative to the control sample by using the $2^{-\Delta\Delta C_T}$ method. The average and standard deviation (SD) of three Cts was calculated, and the average value was accepted if the SD was lower than 0.21 [17]. The N-fold copy number of *Pfsurf*$_{4.1}$ gene between values, 0.86 < N-fold < 1.16, was accepted that the test sample harbored a single copy of the target gene, i.e., N-fold = 1.

## 4. Structure of full-gene *Pfsurf*$_{4.1}$ gene analysis

Sequence comparison of full-length *surf*$_{4.1}$ gene was analyzed using Bioedit software [18] followed with manually edit. To define location of nucleotide sequence between exon1, intron and exon2, obtained 30 complete sequences were aligned with 16 full-gene coding sequences (CDS) and mRNA of reference strains (S1 Table). Nucleotide positions were after 3D7 sequence. All nucleotide sequences were deposited in Genbank database under accession number PP750394- PP750469).

## 5. DNA sequence variation analysis

Tests for departures from neutrality were based on allele frequency distribution indices (Tajima's D, and Fu and Li's D* and F* test) were computed using DNASP6.0 software [19]. Tajima's D test was used to evaluate a departure from the neutral evolution model by comparing θ (nucleotide diversity estimated based on the number of segregating site, S) and π (observed pairwise nucleotide diversity) to investigate whether polymorphic single nucleotide alleles tend to occur at higher or lower frequency than expectation under neutral drift [20]. Fu and Li's D* and F* tests evaluate departures from neutrality by comparing the number of mutations in the external (considered to be "new" mutations) and internal (considered to be "older" mutations) branches of the genealogy. The number of external mutations would be deviated from neutral expectation by the selective pressure, whereas the number of internal mutations is less affected. Under positive balancing selection, the number of internal "old" mutations is expected to be higher than the number of external "new" mutations. Fu and Li's D* compares the estimated θ based on the number of singletons (mutations appearing only once among the sequences, which is new and locates in the external branches) and that based on S. Fu and Li's F* compares the estimated θ based on the number of singletons and that based on k (average number of pairwise nucleotide difference) [21]. Sliding window plots of the nucleotide diversity of 90 bp with step size of 3 bp was analyzed to find region under selection [6, 7]. Statistically significant showed * when P = 0.05, ** when P = 0.02, *** when P = 0.001. Denoted position of nucleotide (nt) and amino acid (aa) are after 3D7 sequence.

## 6. Phylogenetic analysis of SURFIN$_{4.1}$ variants

To find relatedness between SURFIN$_{4.1}$ allelic diversity, evolutionary analyses were conducted in MEGA11 [22]. The obtained 76 *Pfsurf*$_{4.1}$ of this study and deposited 155 *Pfsurf*$_{4.1}$ gene sequences on databases (GenBank and PlasmoDB, data accessed on December 2023), were included in the analysis. Of total 185 sequences, there were 46 full-gene and 139 partial sequences (S1 Table). Evolutionary history was inferred using the Neighbor-Joining method [23]. The tree is drawn to scale, with branch lengths in the same units as those of the evolutionary distances used to infer the phylogenetic tree. The evolutionary distances were computed using the Poisson correction method [24] and are in the units of the number of amino acid substitutions per site. The coding data was translated assuming a standard genetic code table.

All positions containing gaps and missing data were eliminated (complete deletion option). The analysis of coding sequence (exon1-exon2) involved 46 nucleotide sequences and there was a total of 874 positions in the final dataset. The analysis of exon 1 involved 185 amino acid sequences and a total of 685 positions in the final dataset. The analysis of exon2 involved 92 amino acid sequences and there were a total of 86 positions in the final dataset.

## 7. Statistical analysis

Statistical analysis was performed using GraphPad Software, Inc. Accessed 18 March 2024. https://www.graphpad.com/quickcalcs/contingency1/. The difference of data between groups was assessed by nonparametric Fisher's exact test. For all analysis, a P value of less than 0.05 was statistically significant.

## 8. Prediction of transmembrane domain and protein structure

DeepTMHMM (https://dtu.biolib.com/DeepTMHMM) was used to predict cellular protein location [25]. Protein structure was predicted by NetSurfP-3.0 server (https://services.healthtech.dtu.dk/service.php?NetSurfP-3.0) [26].

## 9. Prediction and screening of B cell epitope and helper T lymphocyte (HTL) epitope

B -cell epitope prediction was scanned using BepiPred-3.0 with default setting using sequential smoothing mode (https://services.healthtech.dtu.dk/services/BepiPred-3.0/) [27]. The ABCpred (https://webs.iiitd.edu.in/raghava/abcpred/ABC_submission.html) with default setting of various parameters was employed to predict linear B lymphocyte (LBL) epitope of SURFIN₄.₁ extracellular domain [28, 29]. For mapping potential vaccine candidate, predicted epitopes were evaluated their antigenicity by the VaxiJen v2.0 server (http://www.ddg-pharmfac.net/vaxijen/VaxiJen/VaxiJen.html) based on the basis of physiochemical property of the input peptide [30]. The epitopes with antigenicity scores above threshold 0.5 for parasite model were selected for the assessment of toxicity by (https://webs.iiitd.edu.in/raghava/toxinpred/multi_submit.php) and allergenicity by AllerTOP v.20 server (https://www.ddg-pharmfac.net/AllerTOP/index.html) with default setting [31].

To analyze SURFIN₄.₁ extracellular domain HLA-restricted response, we used the NetMHCII-pan version 4.0 server (https://services.healthtech.dtu.dk/service.php?NetMHCIIpan-4.0), to identify HTL binding epitope, predict peptide length 15 with prediction mode EL, threshold for strong binding peptides (%Rank) 1%, threshold for weak binding peptides (%Rank) 5% [32]. Peptides predicted to bind HLA-Class II, DPB1*02:01, DPB1*04:01, DPB1*05:01, DQA1*01:01, DQA1*01:02, DQA1*03, DQA1*03:01, DQA1*03:01:01, DQA1*03:02, DQA1*05:01, DQB1*02:01, DQB1*03:01, DQB1*03:03:02, DQB1*05:01, DRB1*04, DRB1*07:01, DRB1*09, DRB1*09:01, DRB1*09:01:02, DRB1*12:01, DRB1*15:01, were chosen because of high allele frequency (>0.1) in Thai population [33]. Peptides with strong binding epitope of each allele were included for further analysis. Only The selected epitopes were verified by VaxiJen v2.0, using default threshold value of 0.5 for the prediction. The epitopes with an antigenicity score above 0.5 were screened for *in silico* IFN- inducing using the IFNepitope server, based on support vector machines (SVM) hybrid approach and interleukin-4 (IL4) inducing MHC II binders using IL4pred [34].

## 10. Coverage of predicted immunoepitopes with SURFIN₄.₁ variants

Comparison of amino acid sequence of SURFIN₄.₁ extracellular domain were performed using MEGA11 software [22]. The total of 136 (30 sequences Thai isolates in this study and 106

isolates deposited in GenBank [6, 7, 9] were included for the analysis. The predicted peptides were screen for possible occur in the compared sequences.

## Results

### The organization of full-length *surf$_{4.1}$* gene (exon1-intron-exon2) and genetic variation of coding sequence

*Plasmodium falciparum surf$_{4.1}$* gene composed of two exons (exon1 and exon2) and one intron [4, 8, 10]. From 76 clinical isolates, 30 complete full-length and 46 partial nucleotide sequences of *surf$_{4.1}$* gene were obtained in this study. To define boundary between exon1, intron and exon2, nucleotide sequence comparison of complete gene between obtained 30 full-length and coding sequences (CDS) and mRNA of 16 reference strains was performed. The nucleotide position of exon1, intron and exon2 based on were shown in Fig 1A. The size of *surf$_{4.1}$* gene range between 6753–6769 bp which from exon1 (2367–2373 bp), intron (85–94 bp) and exon2 (4292–4305 bp). Nucleotide (nt) sequence of exon1 (2370 bp) was divided into 3 regions, N-terminal segment (Nter; nt 1–150), cysteine-rich domain (CRD; nt 151–585), variable regions (Var; nt 586–2310 (Var1; nt 586–1506, Var2; 1507–2310) and transmembrane domain (TM; nt 2311–2370). Intron region was between nt 2371–2464. Nucleotide sequence of exon2 was dissected to expanded TM (nt 2465–2538) and three tryptophan-rich domains (WRDs), WRD1 (nt 2539–3946), WRD2 (nt 3947–4558) and WRD3 (nt 4559–6757). Noteworthy, the most frequency of intron nucleotide bases length was 94 bp (93.3%, 28/30), followed by 90 bp (3.3%, 1/30) and 85 bp (3.3%, 1/30). In addition, the feasible splice sites were observed, GT at donor splice site and AG at acceptor splice site after sequence comparison (S1 File). Interestingly, the GT donor splice site was present more than once in the intron in some isolates (13%, 4/30) as well as in reference strains (FCR3, IT).

Sliding window plot examined nucleotide diversity revealed exon1 had higher polymorphic sites while exon 2 showed semiconserved feature as shown in Fig 1B. A total of 447 polymorphic nucleotide sites with maximum of three variants, 2 sites of triplet bases indels, 3 sites of singlet base indels and 2 sites of mini/microsatellites were observed in coding sequence (exon1 and exon2, n = 30) (Table 1). The average pairwise nucleotide diversity was 0.018 for coding sequence. This nucleotide diversity was nearly ten-fold higher in exon1 (0.044) than in exon2 (0.0045). The exon1 showed extensive variation with 385 polymorphic sites and 2 triplet indels sites found. Most of polymorphic sites (371 sites) were in the variable regions (98 sites in Var1 and 273 sites in Var2). While the Nter, CRD and TM had 2, 9 and 3 polymorphic sites, respectively. The two triplet indels (AAT and GGA) were found at nt 1348–1350 and between nt 2268/2269, respectively. These triplet indels located in exon1 could contribute to size variation of SURFIN$_{4.1}$ extracellular region because of in-frame translation to asparagine, (N; AAT) and glycine (G; GGA). For exon2, there were 62 polymorphic nucleotides, one site of triplet base indels, 2 sites of singlet base indels and one site of nine-bases microsatellite found in 30 isolates. When more sequences of exon2 were included (n = 30 to n = 76), the polymorphic nucleotides increased to 70 sites whereas sites of singlet-base remained unchanged. In addition, observed triplet-bases indels were part of three-bases microsatellites. The observed singlet base indels sites were located between nt 2409/2410 and nt 3809/3810. These two sites of singlet base indels were also observed in other strains (3D7, 7G8, KH01, KH02, GB4, HB3, IT, Dd2, GA01, SN01, GN01 and CD01) whereas singlet base indels at nt 4442/4443 was only observed in FCR3 strain. The triplet bases indel between nt 5542/5543 (GAT) was observed in all 76 isolates. Whereas six different of triplet indels, GAT, GAA, GAC, AAT, TAT were observed in other published strains. This difference suggested microsatellite variation in this region causing variation in repeat number of single amino acid. The nine-bases microsatellites found at nt 6149–6134 caused three

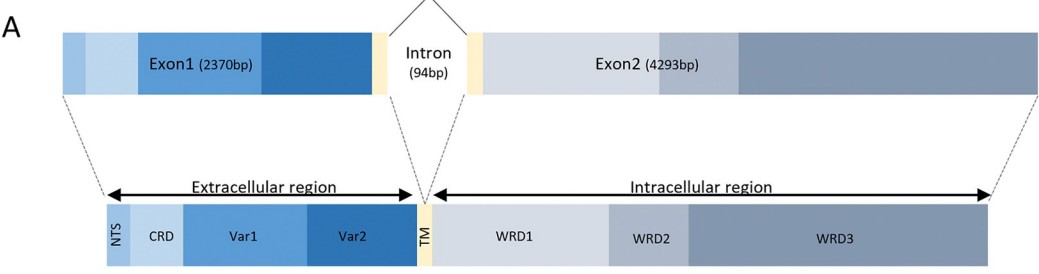

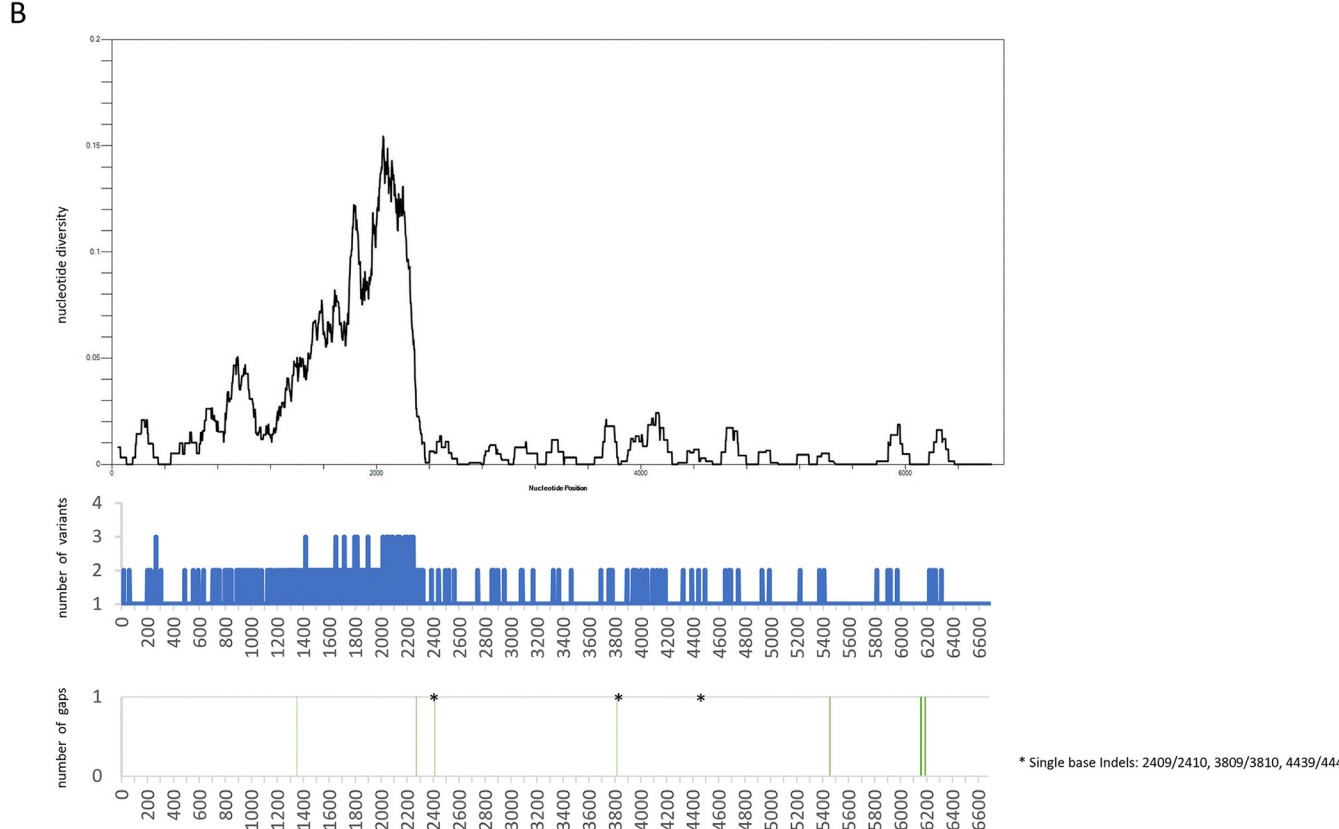

**Fig 1. The organization of *Plasmodium falciparum surf*$_{4.1}$ gene and nucleotide diversity.** (A) Schematic structure of full-length *surf*$_{4.1}$ gene and coding region including the N-terminal sequence (NTS), cysteine-rich domain (CRD), variable regions (Vars), transmembrane region (TM), tryptophan-rich domain (WRD). The length in nucleotide base pair of the reference *P. falciparum* strain 3D7 is shown in parenthesis. (B) Distribution of nucleotide diversity and accumulation of mutation, insertion and deletion (InDels) across *surf*$_{4.1}$ gene. Asterisks (*) indicate positions of single base indels. Nucleotide position is after the *P. falciparum* strain 3D7.

amino acids indels, KNI (`AAAAATATT`), ENI (`GAAAATATT`), GST (`GGAAGCACT`), and GKI (`GGAAAGATT`). These observations suggested minisatellite of three or nine-bases indels caused amino acid indels which contributed to size polymorphism of SURFIN$_{4.1}$.

## Exon1 and exon 2 of the *surf*$_{4.1}$ gene are under selection

Evaluation signature of selection using allele-frequency based indices of balancing selection was performed in full-gene (exon1-exon2) and in each exon. No significant deviation from neutrality was observed when entire sequence of exon1, exon2 and exon1-exon2 were analyzed. To analyze further the region where the potential balancing/directional selection act on

**Table 1. Nucleotide diversity indices of *Plasmodium falciparum* surf$_{4.1}$ from clinical isolates.**

| | Exon1-Exon2 (n = 30) | Exon1 (n = 30) | Exon2 (n = 30) | Exon2 (n = 76) |
|---|---|---|---|---|
| 6660–6675 bp | 1–6679 | 1–2373 | 2374–6679 | 1–4306 |
| Number of sites | 6679 | 2373 | 4306 | 4306 |
| **Total number of sites (excluding sites with gaps / missing data)** | 6650 | 2367 | 4283 | 4276 |
| Sites with alignment gaps or missing data: | 29 | 6 | 23 | 30 |
| Invariable (monomorphic) sites: | 6203 | 1982 | 4221 | 4206 |
| **Number of polymorphic (segregating) sites, S:** | 447 | 385 | 62 | 70 |
| **Total number of mutations, Eta** | 470 | 408 | 62 | 70 |
| Singleton variable sites: | 34 | 29 | 5 | 7 |
| Parsimony informative sites: | 413 | 356 | 57 | 63 |
| Singleton variable sites (two variants) | 34 | 29 | 5 | 7 |
| Parsimony informative sites (two variants) | 390 | 333 | 57 | 63 |
| Parsimony informative sites (three variants) | 23 | 23 | 0 | 0 |
| Number of Haplotypes, h: | 27 | 26 | 26 | 52 |
| Haplotype (gene) diversity, Hd: | 0.993 | 0.991 | 0.991 | 0.986 |
| **Nucleotide diversity, Pi** | 0.01844 | 0.04373 | 0.00445 | 0.00421 |
| **Theta (per site) from Eta** | 0.01784 | 0.04351 | 0.00365 | 0.00334 |
| **Protein Coding Region** | 6678 | 2373 | 4305 | 4305 |
| Total number of sites excluding complex codons or codons with gaps | 6582 | 2313 | 4269 | 4263 |
| Total number of sites in codons with alignment gaps or missing data | 42 | 6 | 36 | 42 |
| Total number of sites in other codons (complex codons) no analyzed | 54 | 54 | na | |
| **Synonymous/Replacement Changes** | | | | |
| Segregating sites | 397 | 335 | 62 | 70 |
| Total number of mutations | 411 | 349 | 62 | 70 |
| Total number of Synonymous changes | 73 | 62 | 11 | 16 |
| Total number of Replacement changes | 338 | 287 | 51 | 54 |

the full-length *surf$_{4.1}$* coding sequence, the sliding window plots of Tajima's D, Fu and Li's D* and F* calculated for *surf4.1* sequences (n = 30) were scanned (Fig 2, Table 2). Significant positive values of Fu and Li's D* were detected at nt 847–1080 (D* = 1.72279, p<0.02), at nt 1270–1605 (D* = 1.68983, p<0.02) and nt 1951–2337 (Fu and Li's D* = 1.83405, p<0.02). These nucleotide positions located in variable regions of exon1 (Var1 and Var2) and N-terminus part of TM region. This result suggested the variable regions of exon1 is under selection. Interestingly, significant positive value of Tajima's D was observed at nt 4089–4241 (D = 2.80755, p<0.01), nt 4593–4745 (D = 2.44751, p<0.05) and nt 6168–6305 (D = 2.1546, p<0.05) of exon2. In addition, the significant positive value of Fu and Li's F* were also detected at nt 4089–4241 (F* = 1.90158, p<0.02) whereas the other two showed high but not statistically significant. When analyzed exon2 of 76 sequences, positive value of Tajima's D remained significant with wider length of nucleotide base. In addition, significant positive value of Fu and Li F* were detect at nt 2220–2396 (D = 2.766, p<0.05). Of note, nt 4089–4241 located in WRD2 while nt 4593–4745 and nt 6168–6305 were in WRD3. The results suggest WRD2 and WRD3 of exon 2 were under diversifying and balancing selections.

## Adenine indels of exon2 cause frameshift mutations that result in different SURFIN$_{4.1}$ variants

Predicted SURFIN$_{4.1}$ protein sequences from 30 nucleotide coding sequences of exon1-exon2 were analyzed in comparison with reference strains (S2 File). The total of 320 amino acid

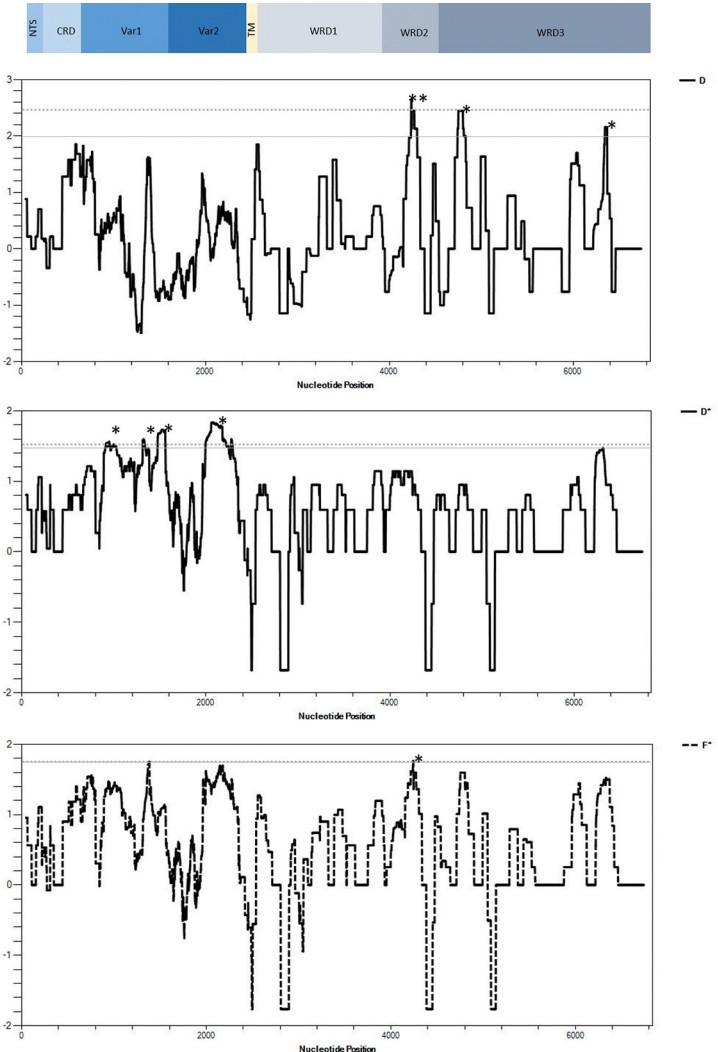

**Fig 2. Test of neutrality for *Plasmodium falciparum surf₄.₁* across 6.7 kb in Thai isolates (n = 30) using a sliding window approach (window size = 90bp, step size = 3bp).** Allele frequency indices (A) Tajima's D, (B) Fu and Li D* and (C) Fu and Li F* were calculated. Sites above the solid line (P<0.05) and broken line (P<0.01 for Tajima's D and P<0.02 for Fu and Li's D* and F*) depart significantly from neutrality (two-tailed), suggesting diversifying selection. Asterisks (*) indicate the region where positive deviation from neutrality were detected in this study. Nucleotide positions are *P. falciparum* strain 3D7.

substitution sites as well as site of each deletion, insertion, mini/microsatellites were found along SURFIN₄.₁ protein sequence as shown in Fig 3A and S2 Table. Of those nonsynonymous substitution sites, 247 and 39 were found in exon1 and exon2, respectively. Of note, residue at 42, 80, 247 and 1122 could found either nonsynonymous or synonymous substitution. The maximum of 5 amino acid residues substitutions observed in the extracellular domain encoded by exon1. Two amino acid indels (N450 and G756/757) were also found in the variable domain of exon1. Only minisatellite of single amino acid (D1797) was observed in exon2 of all isolates in this study. Unlike in published sequences where singlet amino acid D, E, N, Y could be found. Triplet amino acid minisatellites (ENI, KNI, GST, GKI) were found between aa 2032–2046. Of note, both single and triplet minisatellites were in WRD3 of exon2.

Interestingly, the singlet indels of adenine base found between nt 2503/2504, nt 3903, and nt 4536/4537 in exon2 caused frameshift translation in the C-terminal part of SURFIN₄.₁.

Table 2. Test of neutrality for *Plasmodium falciparum surf*$_{4.1}$ from clinical isolates.

| | Nucleotide position | Number of sites (base) | Total number of sites (excluding sites with gaps) | Indels sites | Total number of InDels events | Number of Haplotypes (h) | Nucleotide diversity (per site) (Pi) | Theta | Average number of nucleotide differences (k) | Total number of mutations (Eta) | Number of variable sites (S) | Number of singleton sites | Two variants singleton | Two variants non singleton | >2 variants non singleton | Tajima's D | Fu and Li's D* | Fu and Li's F* |
|---|---|---|---|---|---|---|---|---|---|---|---|---|---|---|---|---|---|---|
| E1-E2 (n = 30) | 1–6786 | 6786 | 6735 | 28 | 14 | 27 | 0.019 | 0.018 | 126.609 | 486 | 462 | 34 | 34 | 404 | 24 | 0.125 | 1.395# | 1.090 |
| E1 (n = 30) | 2373 | 2373 | 2367 | 6 | 2 | 26 | 0.044 | 0.044 | 103.520 | 408 | 385 | 29 | 29 | 333 | 23 | 0.020 | 1.373# | 1.034 |
| | 847–1080 | 234 | 234 | 0 | 0 | 13 | 0.036 | 0.031 | 8.529 | 29 | 29 | 0 | 0 | 29 | 0 | 0.595 | 1.722** | 1.483# |
| | 1270–1506 | 237 | 234 | 3 | 1 | 14 | 0.042 | 0.047 | 9.807 | 44 | 43 | 1 | 1 | 41 | 1 | -0.433 | 1.634** | 1.031 |
| | 1507–1605 | 99 | 99 | 0 | 0 | 4 | 0.063 | 0.079 | 6.246 | 31 | 31 | 1 | 1 | 30 | 0 | -0.731 | 1.523* | 0.865 |
| | 1951–2043 | 93 | 93 | 0 | 0 | 20 | 0.125 | 0.100 | 11.625 | 37 | 33 | 0 | 0 | 29 | 4 | 0.897 | 1.723** | 1.667# |
| | 2173–2337 | 165 | 162 | 3 | 1 | 13 | 0.072 | 0.064 | 11.674 | 41 | 37 | 1 | 1 | 32 | 4 | 0.472 | 1.548** | 1.367# |
| E2 (n = 30) | 1–4307 | 4307 | 4282 | 14 | 4 | 26 | 0.004 | 0.004 | 19.078 | 62 | 62 | 5 | 5 | 57 | 0 | 0.824 | 1.295# | 1.239 |
| | 1716–1868 | 153 | 153 | 0 | 0 | 3 | 0.017 | 0.008 | 2.561 | 5 | 5 | 0 | 0 | 5 | 0 | 2.806** | 1.144 | 1.766** |
| | 2221–2373 | 153 | 153 | 0 | 0 | 2 | 0.010 | 0.005 | 1.545 | 3 | 3 | 0 | 0 | 3 | 0 | 2.4474* | 0.950 | 1.486# |
| | 3797–3934 | 138 | 129 | 9 | 1 | 3 | 0.011 | 0.006 | 1.451 | 3 | 3 | 0 | 0 | 3 | 0 | 2.155* | 0.950 | 1.395# |
| E2 (n = 76) | 1–4307 | 4307 | 4276 | 30 | 7 | 52 | 0.004 | 0.003 | 18.006 | 70 | 70 | 7 | 7 | 63 | 0 | 0.870 | 1.139 | 1.179 |
| | 1602–1868 | 267 | 267 | 0 | 0 | 13 | 0.012 | 0.007 | 3.300 | 9 | 9 | 0 | 0 | 9 | 0 | 2.076* | 1.334# | 1.791* |
| | 2220–2396 | 177 | 177 | 0 | 0 | 5 | 0.011 | 0.005 | 1.899 | 4 | 4 | 0 | 0 | 4 | 0 | 2.766** | 0.965 | 1.722* |
| | 3805–3933 | 129 | 120 | 9 | 1 | 4 | 0.012 | 0.005 | 1.410 | 3 | 3 | 0 | 0 | 3 | 0 | 2.463* | 0.851 | 1.513# |

A

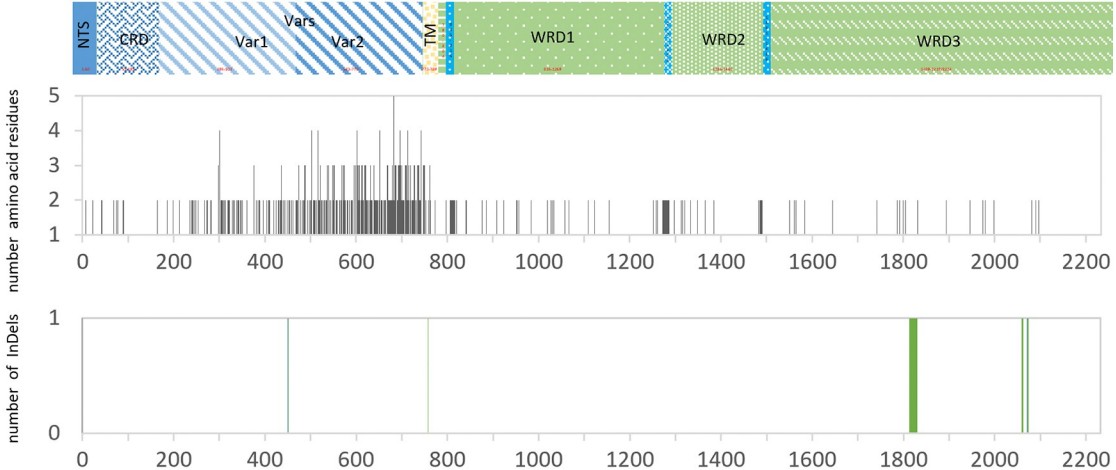

B

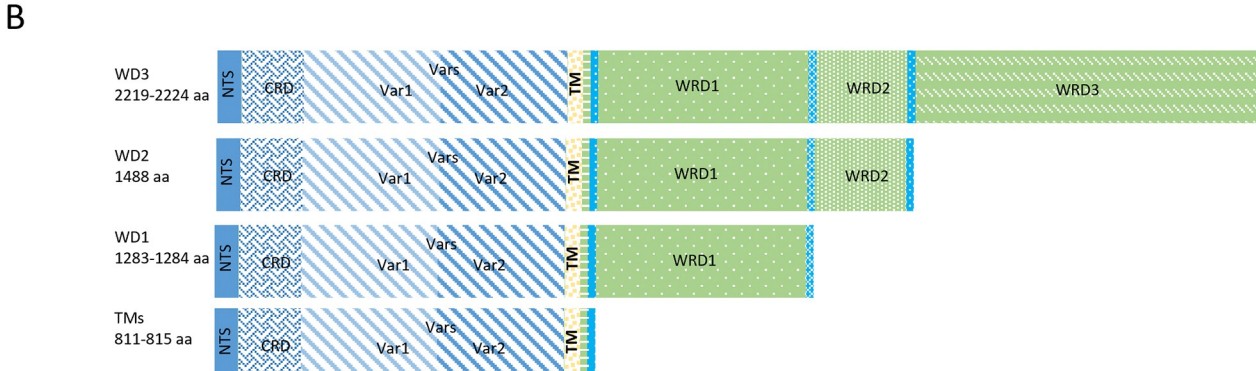

**Fig 3. The amino acid sequence of SURFIN$_{4.1}$ and type of variants.** (A) Schematic structure of SURFIN$_{4.1}$ including the N-terminal sequence (NTS), cysteine-rich domain (CRD), variable regions (Vars), transmembrane region (TM), tryptophan-rich domain (WRD). Accumulation of amino acid residue substitution, insertion and deletion (InDels) across the SURFIN$_{4.1}$; the length in amino acids of the reference *P. falciparum* strain 3D7 is presented in parenthesis. (B) Different types of SURFIN$_{4.1}$ variants; transmembrane variants (TMs), WD1 variant containing one WRD (WRD1), WD2 variant containing two WRDs (WRD1 and WRD2), and WD3 variant containing three WRDs (WRD1, WRD2 and WRD3). The length of amino acids shown in parenthesis is after the reference *P. falciparum* strain 3D7.

Truncated SURFIN$_{4.1}$ lacking WRDs had no alanine insertion at 2503/2504 which translated in-frame with three following stop codons (TGA, TAA, TAA). Whereas alanine insertion at this position lead to frameshift mutation that resulted in readthrough mutation which permits translation of full-length SURFIN$_{4.1}$ with three WRDs. After readthrough, additional alanine insertion at 3903 or at 4536/4537 causing SURFIN$_{4.1}$ with only one WRD or two WRDs, respectively. The outcome of alanine base insertion at key positions in the C-terminal caused truncation of transmembrane part of SURFIN$_{4.1}$ produced different form of SURFIN$_{4.1}$, transmembrane (TM) variant, types contained one WRD (WD1 variant), two WRDs (WD2 variant) and three WRDs (WD3 variant) as illustrated in Fig 3B (S2 Table). The TM variant was further divided into two subgroups, TM1 and TM2, as a point mutation in first stop codon (TGA> AGA), resulted in readthrough translation adding three more amino acid residues.

Of those 30 isolates, the TM variants was the most observed with frequency 63% (TM1; 11/30, TM2; 9/30), while the WDs variants was found with frequency 37% (WD3; 10/30, WD1; 1/30). Noteworthy, the WD2 was not observed in this study and only found in FCR3 reference

strain. In addition, there were 44 tryptophan coding nucleotides (TGG) in total found in exon2. Of these, 14, 12 and 18 were in WRD1 (W1-14), WRD2 (W15-26) and WRD3 (W27-44), respectively. Interestingly, all except one (W22) of these tryptophan coding nucleotides were conserved. The third-base mutation at 4149 cause nonsynonymous mutation, TGG > TGC, changes from tryptophan to cysteine (C). This mutation also found on 7G8, HB3, SD01 reference strains. This result suggested combination of adenine indels at three different positions in exon2 caused frameshift mutation resulted in polymorphic forms of SURFIN_{4.1}.

## The copy number variation of *surf_{4.1}* gene

The copy number variation (CNV) of *surf_{4.1}* gene was investigated. The total 70 of 76 isolates were successfully examined gene copy number. Copy number of one (1CNV) were the most observed with frequency of 87% (61/70) whilst more than one copy ($\geq$2CNV) was detected with frequency of 13% (9/70). The CNV observed were 18, 5, 3, and 2 found with frequency of 1% (1/70), 1% (1/70), 3% (2/70) and 7% (5/70), respectively (S3 File). The distribution of *surf_{4.1}* CNV with type of SURFIN_{4.1} variants was investigated. The TM variants with 1CNV and with $\geq$2CNV were found with frequency of 89.6% (43/48) and 10.4% (5/48), respectively. Whereas WDs variant with 1CNV and $\geq$2CNV were found 81.8% (61/70) and 12.9% (9/70), respectively. there was no significant difference between *surf_{4.1}* CNV and type of SURFINs variants (p = 0.4350, Fisher's exact, 2 tailed) (S3 Table).

## Clinical outcomes, SURFIN_{4.1} variants and CNV

The frequency of SURFIN_{4.1} variants in two clinical outcomes were investigated. The total of 76 sequences of exon2 were obtained from clinical isolates (23 and 53 of mild and complicated malaria, respectively). SURFIN_{4.1} variant TMs was detected with frequency of 67% (51/76) and variants WDs with 33% (25/76) frequency. In mild malaria, there were found with frequency of 52.2% (1223) and 47.8% (11/23) of TM and WDs variants, respectively. While in complicated malaria, TMs and WDs variants were found at frequency of 73.6% (39/53) and 26.4% (14/53), respectively. The frequency of WDs variants was lower in complicated than in mild malaria. However, this difference was not statistically significant (p = 0.1096, Fisher's exact test, two-tailed) (S3 Table).

The distribution *surf_{4.1}* CNV in two clinical outcomes was further analyzed. In mild malaria, there were 90.9% (20/22) and 9.1% (2/22) of 1CNV and $\geq$2CNV found, respectively. While in complicated malaria, 1CNV and $\geq$2CNV were found at frequency of 85.4% (41/48) and 14.5% (7/48), respectively. There was no significant difference between CNV and clinical outcomes (p = 0.7094, Fisher's exact test, two-tailed) (S3 Table).

## The relatedness of SURFIN_{4.1} variants

SURFIN_{4.1} showed extensive diversity in both genetics and protein levels. The total of 26 haplotypes of 30 *surf_{4.1}* nucleotides sequences were observed with the highest frequency of 6.7% (2/30) as well as SURFIN_{4.1} showing multiple type variants. The population structure of SURFIN_{4.1} was further analyzed to find relatedness within *P. falciparum* population. The SURFIN_{4.1} complete nucleotide coding sequence (exon1 and exon2) was grouped into 5 distinct clades, clade 1 FCR3/IT/SD01/SN01/KH01/TH (this study, 11 isolates), clade 2 GA01/KE01/KH02/GN01/TH (this study, 5 isolates), clade 3 3D7/MS822/Dd2/GB4/TH (this study, 6 isolates), clade 4 TH (this study, 3 isolates) and clade 5 7G8/CD01/HB3/TH (this study, 5 isolates) as shown in Fig 4A. Noteworthy, the WDs and TM variants were both found in four clades,

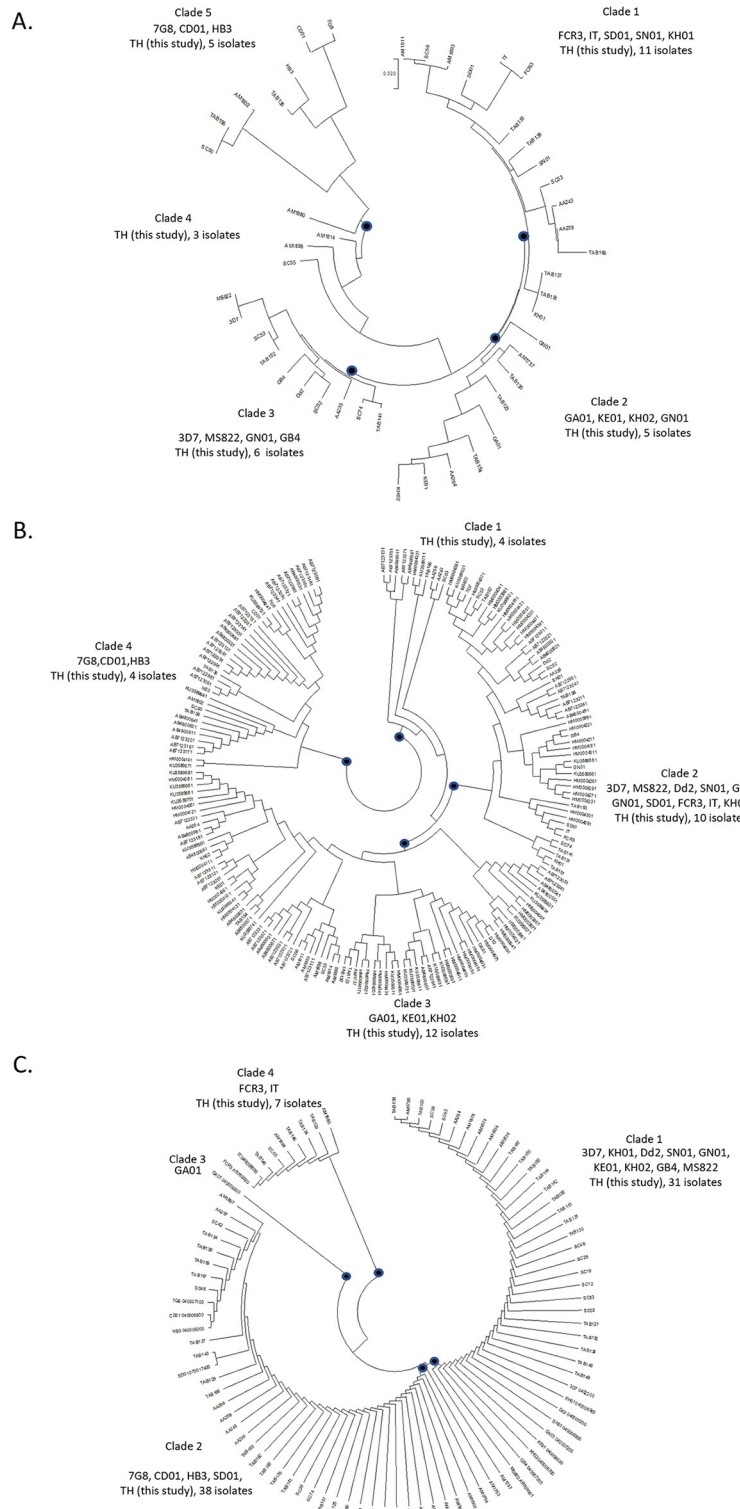

**Fig 4. Phylogenetic tree for *P. falciparum* isolates.** Constructed using (A) coding nucleotide sequence (exon1 and exon2, n = 46) (B) exon 1(n = 185), and (C) exon 2 (n = 92) of *surf*$_{4.1}$.

clade 1 (WD3/WD2/TM1/TM2), clade 2 (WD3/TM1), clade 4 (WD3/TM2), clade 5 (WD3/WD1).Whereas clade 3 was found only TM variants (TM1/TM2).

When analyzed exon1 and exon2 separately, there were only 4 clades grouped for both exons, exon 1: clade 1 TH (this study, 4 isolates), clade 2 3D7/MS822/Dd2/GB4/ FCR3/IT/SD01/KH01/SN01/GN01/TH (this study, 10 isolates), clade 3 GA01/KE01/KH02/TH (this study, 12 isolates) and clade 4 7G8/CD01/HB3/TH (this study 4 isolates), exon2: clade 1 3D7/MS822/Dd2/KH01/SN01/GN01/KE01/KH02/GB4/TH (this study, 31 isolates), clade 2 7G8/CD01/HB3/SD01/TH (this study, 38 isolates), clade 3 GA01 and clade 4 FCR3/IT/TH (this study, 7 isolates) (Fig 4B and 4C). Of note for exon2, clades 1 and 3 were found only TM variants while mixed TMs and WDs variants were observed in both clades 2 and 4. Likewise for exon 1, TM variants only found in one clade (clade 1) whereases mixed variants (TMs and WDs) were observed in three clades (clades 2–4). These results suggesting a closely related of SURFIN4.1 variants, especially the extracellular domain which encoded by exon1. Noteworthy, reference strains including in the analysis were isolated from different continents. Hence this also indicating SURFIN4.1 variants were conserved across continents.

## Prediction of transmembrane and secondary protein structure

Predicted domain location of 4 isoforms of SURFIN4.1 were analyzed by DeepTMHMM using *Pfsurf4.1* nucleotide sequences obtained in this study. Transmembrane topology prediction showed amino acid position 1-770/778 located outside, transmembrane motifs (VPVALAVFGVLFVFILF, aa 771–787 for TM1, TM2, WD1 and FGVLFVFILF, aa 779–788 for WD2 and WD3) and amino acid located inside (TM1; 788–811, TM2; 788–814, WD1; 788–1283, WD2; 789–1488, and WD3; 789–2224) were identified as shown in Fig 5. Specific amino acid residues at C-terminal of each type were observed, GKSDEEEGMM in TM1, GKSDEEGMMRKY in TM2, KQMENRYRNIYGSDE in WD1, CSNIYR in WD2 and 130 conserved residues in WD3 types (S4 Table).

Predicted secondary structure of SURFIN4.1 variants revealed 5 helix domains located between aa 20–175 of extracellular domain in all isoforms (Fig 6). The variable regions showed mainly coils intersperse by few strands and small helix turn. Intracellular domains showed coils in the N and C terminal and intersperse with 15 helix domains (5 in WRD1, 4 in WRD2, and 6 in WRD3). SURFIN4.1 TMs type contained no helix domain while WD1, WD2 and WD3 had 5, 9 and 15 helix domains, respectively.

## Prediction and analysis SURFIN4.1 B and T helper cells epitopes

Humoral immune response to malaria infection is a protective immunity. Protein expressed on the surface of infected red cells can trigger immune response are targeted for malaria vaccine development. The extracellular domains of 4 clades SURFIN4.1 were scanned for B cell epitope prediction by BepiPred-3.0 as showed in Fig 7A. Interestingly, extracellular domain of all clades showed similar predicted pattern of B cells epitope location. Amino acids with predictive score higher than default score 0.1512 were residues 205–351 and 434–494 which located in variable region. The peptide sequences of predicted B cell epitopes were illustrated in Fig 7B. This result suggested feasibility to design immunogenic peptides of SURFIN4.1 for all clades.

To further analysis, an isolate of clade 4 was selected as a represented SURFIN4.1 to predicted define epitopes for B-cell and helper T-cell epitopes. As well as testing for antigenicity, allergenicity and toxicity as qualified immunogenic peptides. The extracellular domain was found to be antigenic with value 0.6596 and 0.6680 without (aa 1–778) and with transmembrane (aa 1–788) residues, respectively, as estimated by vaxijen2.0 webserver. Initially, there

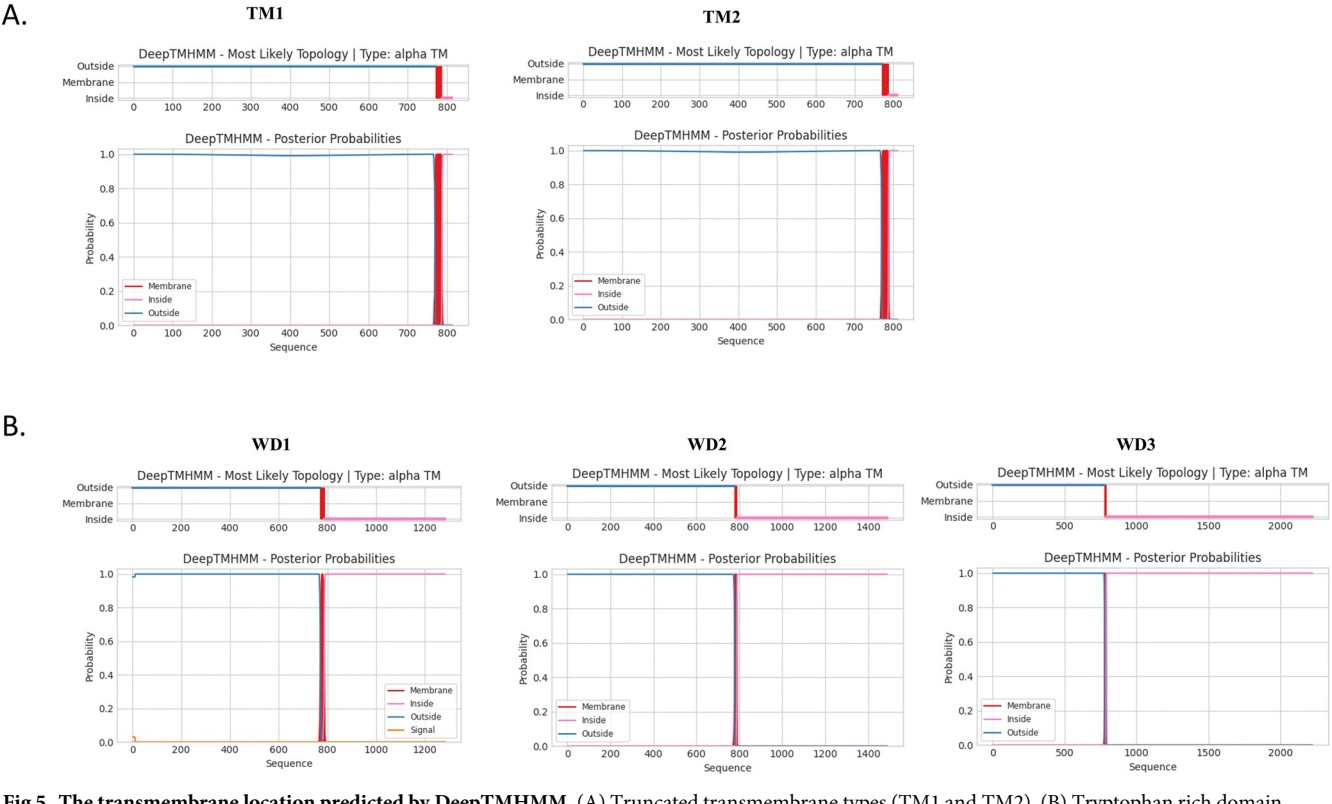

**Fig 5. The transmembrane location predicted by DeepTMHMM.** (A) Truncated transmembrane types (TM1 and TM2). (B) Tryptophan rich domain types (WD1-3).

were 83 Linear B lymphocyte (LBL) epitopes found with scores greater than 0.51, indicative of active B cell immune response. Predicted LBL epitopes were testing for antigenicity, allergenicity and toxicity, there were 35 Linear B lymphocyte after the screening (Table 3). Of those, 13 epitopes were located between residues 205–351 and 434–494, as predicted by BepiPred-3.0. Moreover, 79 peptides were predicted as HTL strong binding epitope using NetMHCIIpan server. After screening for antigenicity, allergenicity and toxicity, 37 HTL epitopes were tested for IFN-γ and IL-4 inducer to find better epitope (Table 3). There were 13 peptides unable to induce neither IL-4 nor FN-γ. Whereas 4 peptides were able to induce IFN-γ, 24 peptides were able to induce IL-4 and 4 peptides were able induce both IFN-γ and IL-4. Interestingly, 3 peptides (2, 174, and 764) were both LBL and HTL binding epitope. Peptides (56, 107, 272, 334, and 746) showed overlapped residues of LBL and HTL. Similarity of predicted peptides compared with 136 isolates (n = 136, 30 isolates in this study and 106 isolates of SURFIN$_{4.1}$ deposited in GenBank) were explored. Despite highly polymorphic of the extracellular domain, conserved residues of LBL epitopes were observed in 4 peptides (45, 107, 123 and 134) which located in Nterm and CRD. In addition, dimorphic peptides (2, 174, 205 and 211) were identified which located in Nterm, CRD and the variable regions.

## Discussion

SURFINs protein family present on surface of infected red blood cells and on merozoite surface making them an important target for erythrocytic stage infection. Also, several studies reported naturally acquired antibody against SURFINs members developed during malaria natural infection, indicating as important target of protective immunity [5, 35]. In addition,

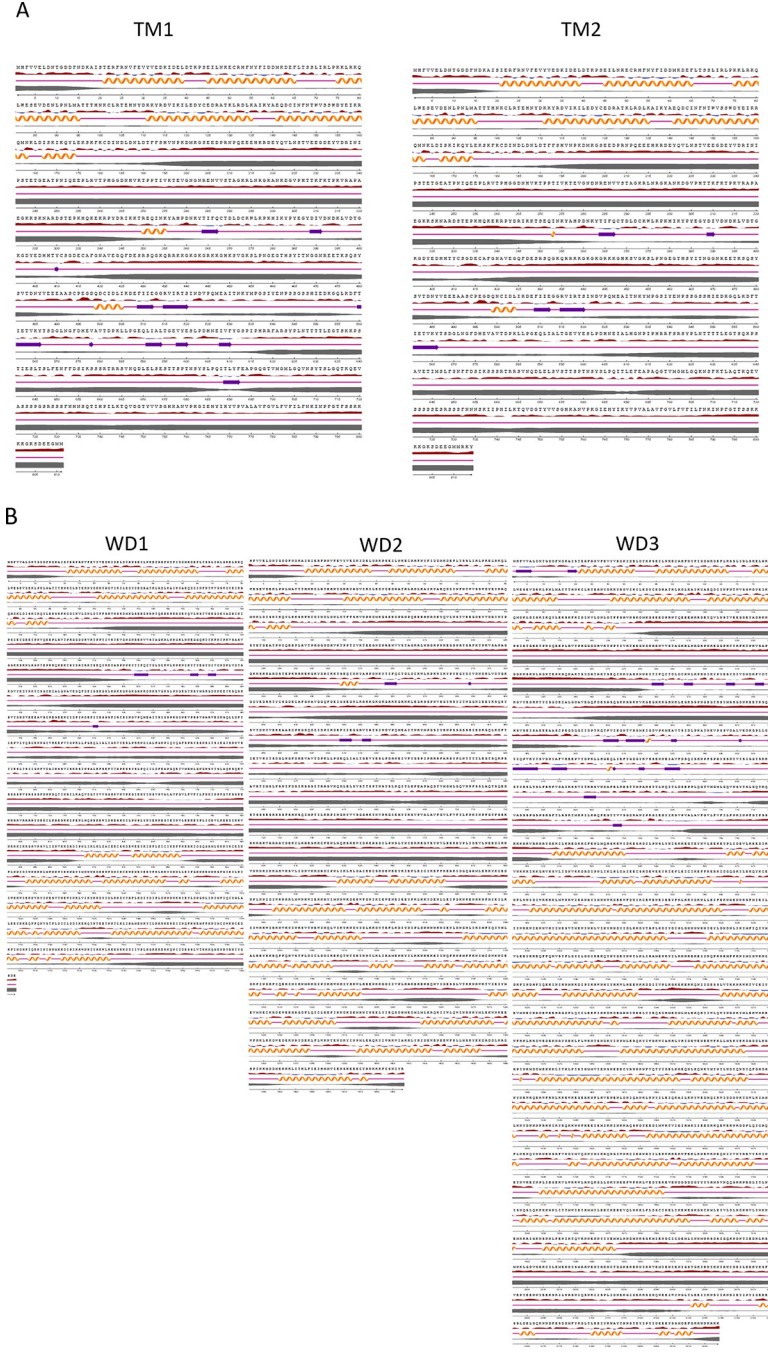

**Fig 6. Secondary structure of SURFIN$_{4.1}$ different variants.** (A) Truncated transmembrane types (TM1 and TM2). (B) Tryptophan rich domain types (WD1-3). Relative surface accessibility showed exposed in red and buried in blue at threshold 25%; secondary structure showed helix, strand and coil; disorder represented by the thickness of line equals probability of disordered residue.

consistent expression of SURFINs in blood stage forms indicates biological role of the protein [10, 36]. These findings supporting important role of SURFINs as a candidate for malaria vaccine development.

A

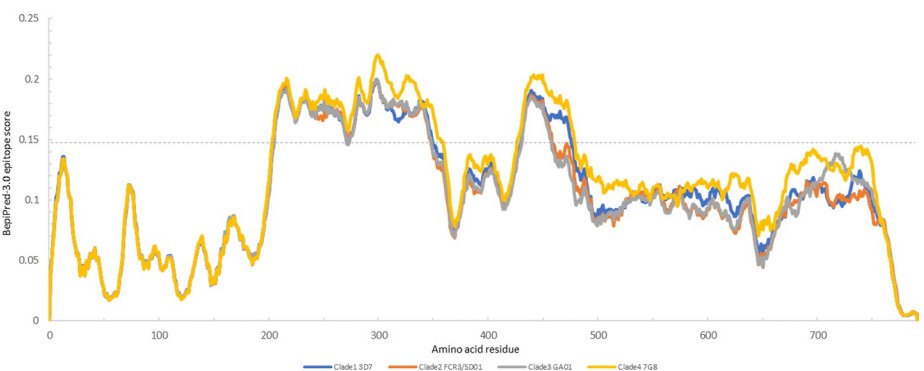

B

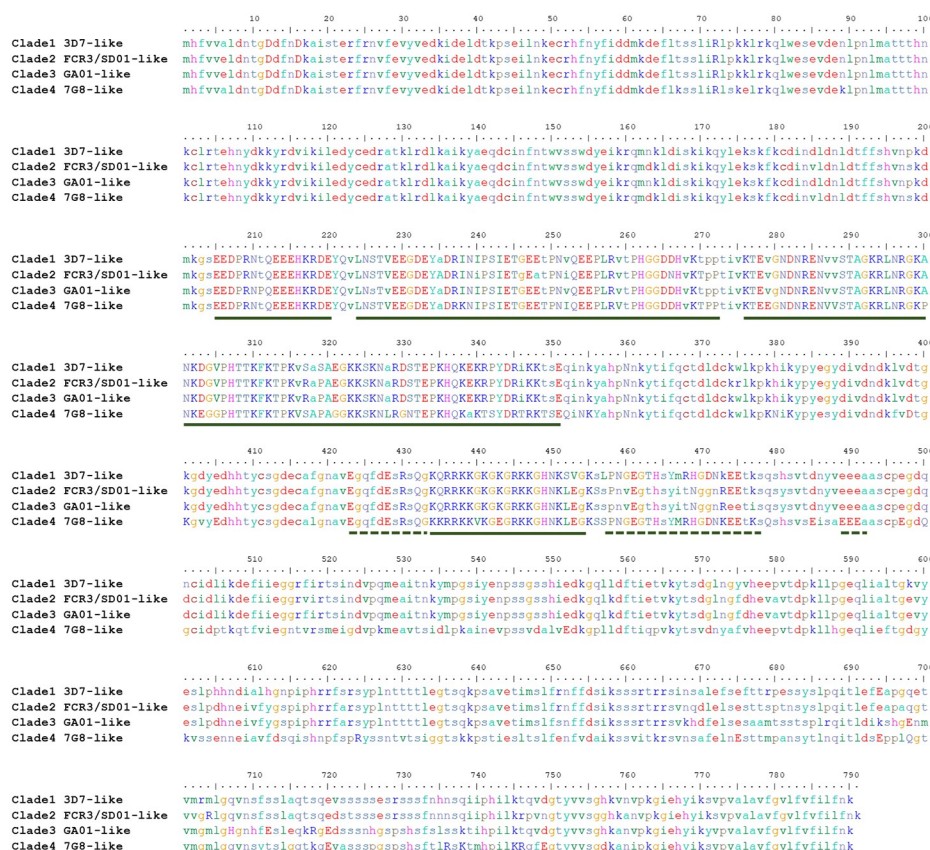

**Fig 7. B cell epitope prediction of extracellular domain of SURFIN₄.₁.** (A) BepiPred-V3.0 predicting B-cell epitope of extracellular domain (amino acids of exon 1, aa1-790) using BepiPred-v3.0, default is 0.1512. (B) Identify B-cell epitope of four clades was underlined.

**Table 3. Predicted B cell and MHC class II epitopes of SURFIN$_{4.1}$ extracellular domain (1–778) and simulate IL 4 and IFNγ inducer.**

| Peptide ID | B cell binding peptides | HTL binding peptides (Thai alleles) | IL-4 | IFN-γ |
|---|---|---|---|---|
| 2 | HFVVALDNTGDDFNDK | HFVVALDNTGDDFND | IL4-inducer | Negative |
| 14 | | FNDKAISTERFRNVF | IL4-inducer | Positive |
| 15 | | NDKAISTERFRNVFE | IL4-inducer | Positive |
| 21 | | TERFRNVFEVYVEDK | IL4-inducer | Negative |
| 45 | EILNKECRHFNYFIDD | | | |
| 56 | YFIDDMKDEFLKSSLI | | | |
| 62 | | KDEFLKSSLIRLSKE | IL4-inducer | Negative |
| 63 | | DEFLKSSLIRLSKEL | IL4-inducer | Negative |
| 65 | | FLKSSLIRLSKELRK | Non-IL4-inducer | Negative |
| 66 | | LKSSLIRLSKELRKQ | Non-IL4-inducer | Negative |
| 67 | | KSSLIRLSKELRKQL | IL4-inducer | Negative |
| 68 | SSLIRLSKELRKQLWE | | | |
| 107 | HNYDKKYRDVIKILED | | | |
| 108 | | NYDKKYRDVIKILED | IL4-inducer | Negative |
| 123 | YCEDRATKLRDLKAIK | | | |
| 125 | | EDRATKLRDLKAIKY | Non-IL4-inducer | Negative |
| 126 | | DRATKLRDLKAIKYA | Non-IL4-inducer | Negative |
| 127 | | RATKLRDLKAIKYAE | Non-IL4-inducer | Negative |
| 128 | | ATKLRDLKAIKYAEQ | Non-IL4-inducer | Negative |
| 129 | | TKLRDLKAIKYAEQD | Non-IL4-inducer | Negative |
| 130 | | KLRDLKAIKYAEQDC | Non-IL4-inducer | Negative |
| 134 | LKAIKYAEQDCINFNT | | | |
| 174 | LEKSKFKCDINVLDNL | LEKSKFKCDINVLDN | IL4-inducer | Negative |
| 188 | | NLDTFFSHVNSKDMK | IL4-inducer | Negative |
| 189 | | LDTFFSHVNSKDMKG | IL4-inducer | Negative |
| 190 | | DTFFSHVNSKDMKGS | Non-IL4-inducer | Negative |
| 205 | EEDPRNTQEEEHKRDE | | | |
| 211 | TQEEEHKRDEYQVLNS | | | |
| 217 | | KRDEYQVLNSTVEEG | Non-IL4-inducer | Negative |
| 218 | | RDEYQVLNSTVEEGD | Non-IL4-inducer | Negative |
| 225 | NSTVEEGDEYADRKNI | | | |
| 272 | PTIVKTEEGNDNRENV | | | |
| 281 | | NDNRENVVSTAGKRL | Non-IL4-inducer | Negative |
| 282 | | DNRENVVSTAGKRLN | Non-IL4-inducer | Negative |
| 283 | | NRENVVSTAGKRLNR | IL4-inducer | Negative |
| 284 | | RENVVSTAGKRLNRG | IL4-inducer | Negative |
| 285 | | ENVVSTAGKRLNRGK | IL4-inducer | Negative |
| 291 | AGKRLNRGKPNKEGGP | | | |
| 299 | KPNKEGGPHTTKFKTP | | | |
| 305 | GPHTTKFKTPKVSAPA | | | |
| 313 | TPKVSAPAGGKKSKNL | | | |
| 334 | PKHQKAKTSYDRTRKT | | | |
| 345 | | RTRKTSEQINKYAHP | IL4-inducer | Positive |
| 413 | GDECALGNAVEGQFDE | | | |
| 421 | AVEGQFDESRSQGKKR | | | |
| 428 | ESRSQGKKRRKKVKGE | | | |

*(Continued)*

**Table 3.** (Continued)

| Peptide ID | B cell binding peptides | HTL binding peptides (Thai alleles) | IL-4 | IFN-γ |
|---|---|---|---|---|
| 435 | KRRKKVKGEGRKKGHN | | | |
| 441 | KGEGRKKGHNKLEGKS | | | |
| 449 | HNKLEGKSSPNGEGTH | | | |
| 489 | EEEAASCPEGDQGCID | | | |
| 495 | CPEGDQGCIDPTKQTF | | | |
| 501 | GCIDPTKQTFVIEGNT | | | |
| 549 | ALVEDKGPLLDFTIQP | | | |
| 555 | GPLLDFTIQPVKYTSV | | | |
| 605 | | ENNEIAVFDSQISHN | IL4-inducer | Negative |
| 621 | FSPRYSSNTVTSIGGT | | | |
| 651 | | ENFVDAIKSSVITKR | IL4-inducer | Negative |
| 685 | TLNQITLDSEPPLQGT | | | |
| 706 | | GQVNSYTSLGQTKQE | IL4-inducer | Positive |
| 712 | TSLGQTKQEVASSSPG | | | |
| 720 | EVASSSPGSPSHSFTL | | | |
| 744 | | LKRQFEGTYVVSGDK | IL4-inducer | Negative |
| 745 | | KRQFEGTYVVSGDKA | IL4-inducer | Negative |
| 746 | RQFEGTYVVSGDKANI | | | |
| 749 | | EGTYVVSGDKANIPK | IL4-inducer | Negative |
| 764 | GIEHYIK<u>S</u>VPVALAVF | GIEHYIKSVPVALAV | IL4-inducer | Negative |
| 765 | | IEHYIKSVPVALAVF | IL4-inducer | Negative |
| 766 | | EHYIKSVPVALAVFG | IL4-inducer | Negative |

The extracellular part of SURFIN$_{4.1}$ is composed of the N-terminal cysteine rich domain (CRD), variable regions (Vars) and transmembrane region (TM). The intracellular part is the continuing of transmembrane domain and contains several tryptophan rich domains (WRDs) at the C-terminal part [4]. The *surf$_{4.1}$* gene is composed of two exons and one intron located on chromosome 4 of *P. falciparum*. A few studies reported extensive sequence variation of the variable regions of the N-terminal part [6, 7, 9], whereas the extent of genetic variation of exon 2 is undefined. Moreover, truncated SURFIN$_{4.1}$ lacking WRD or having different number of WRDs was observed and suggesting differential location of expression [7, 8].

We studied genetic variation of *surf$_{4.1}$* gene as well as CNV of *P. falciparum* population obtained from clinical isolates. The diversity of SURFIN$_{4.1}$ variants and allelic relatedness were analyzed to determine immunopeptides as a vaccine candidate.

Nucleotide diversity of *surf$_{4.1}$* gene is higher in exon1 than in exon2, which encoding extracellular and intracellular domain, respectively. Extensive genetic variation observed on the variable regions of exon1 with significant signature of selection detected was similar with previously studies of Thai and Kenya *P. falciparum* populations [6, 7, 9]. The selection pressure is likely to be the host immune system as the protein located on the merozoite surface and surface of infected red blood cell [8, 10]. The N-terminal part of SURFIN$_{4.1}$ is required for its trafficking to the red blood cell cytosol through the endoplasmic reticulum [8] with minimum five charged amino acids (E38, K42, E45, K49, and E50) are important for efficient export to the red blood cell [37]. In addition, this study was also found TM region under selection. Since TM region is essential for the SURFIN$_{4.2}$ trafficking to the iRBC and Maurer's clefs [38]. Deduce similar function of the region, mutation found in the TM might affect the trafficking of SURFIN$_{4.1}$ is needed to be verified.

Surprisingly, the semiconserved feature of exon 2 which located intracellular also displayed genetic variation and signature of selections were also detected in the WRD2 and WRD3. The plausible explanations might be function of protein domains or immune evasion as extensive genetic variation observed on the extracellular domain. The main function of WRDs of SURFIN$_{4.1}$ has not been verified yet. However, the WRDs of SURFIN$_{4.2}$ were found to directly interact with inside-out vesicles (IOVs) of RBC, especially WRD2 of SURFIN$_{4.2}$ bound to F-actin, the RBC membrane skeleton. This suggested the interaction between WRD and RBC membrane skeleton might be a common feature of WRD-containing proteins [39]. The RBC membrane skeleton composed of several molecules, spectrin, actin, band 3. It is plausible that the variation observed in WRD2 and WRD3 might be due to multiple domain binding during exporting to RBC membrane surface.

Based on cDNA sequencing of *surf$_{4.1}$* transcripts, truncated SURFIN$_{4.1}$ without WRD or with various number of WRDs at the C-terminal part were observed in different strains, suggesting frameshift mutation caused premature stop codon [8]. This different allelic type was reported in western Kenya population by analysis of stop codon present in genomic DNA sequence [7]. In this study, sequence comparison analysis of complete sequence of *surf$_{4.1}$* gene obtained in this study, mRNA, cDNA and DNA sequence of reference strains deposited in database enable the identification of adenine indels present at key positions in exon 2 were likely the causative of frameshift mutation producing different SURFIN$_{4.1}$ variants. Likely, if no adenine insertion at nucleotide 2503/2504 leading to premature stop codon during transcription producing TM variants. While single adenine insertion at this position is essential to produce WDs variants. In combination with additional second adenine insertion at nucleotide 3903, could lead to producing different WDs variants. These findings indicate SURFIN$_{4.1}$ variants caused by frameshift mutation of adenine indels in coding sequence. Implication of this modulation of orientational expressed SURFIN$_{4.1}$ is not well understood. This might suggest that WRDs may be important for the translocation of the protein from Maurer's clefts to the surface of infected red blood cell. This implied expressed SURFIN$_{4.1}$ TM variant present on merozoite surface while SURFIN$_{4.1}$ WDs variants could express on both merozoite and the infected red blood cell surface. The observed frequency of SURFIN$_{4.1}$ WDs variant (33%) from this study was like those (39%) reported study of Thai isolates [6].

Copy number variations (CNVs) are the target of selection in natural populations of *P. falciparum* [40]. Several studies suggested role of CNV in providing adaptability to parasites to the environments such as in adapted to in vitro culture conditions [41], and drug resistance [42]. In this study, the CNV of *surf$_{4.1}$* was detected in clinical isolates. This finding is similar with previous studies observed CNV in different parasite laboratory strains and isolates from western Kenya lines [7, 10]. However, previous study reported no CNV detected in Thai population [7]. Since frequency of observed CNV was around 13%, the inconsistent finding might be limited number of isolates tested from previous study.

In this study, CNV was found to be no different either between clinical outcomes or type of SURFINs. The increase in copy number of *surf$_{4.1}$* gene correlated with increase in RNA transcription, however, the increase in copy number of the gene was not reflected on level of protein [10]. SURFIN$_{4.1}$ was detected on merozoite surface in late schizont stage, a stage before merozoite egress from red blood cell, suggested *surf$_{4.1}$* transcripts are under controlled in a highly time-dependent manners. In accordance with the finding of PfAlba1, a post-transcriptional regulator, binds to *surf$_{4.1}$* transcripts among with other knowns erythrocyte invasion components [43]. These finding supports translational repression acts on the level of SURFIN$_{4.1}$ rather than number of gene copy. Hence, the benefits of *surf$_{4.1}$* CNV remain to be elucidated.

SURFINs possess components of two major virulence factors, N-terminal contains cysteine rich domain (CRD) and tryptophan rich domains (WRDs) at the C-terminal part similar to PfEMP1 [4]. Function of PfEMP1 protein is to mediate cytoadhesion of iRBC to host cells. Sequestration is the biding of iRBC to host receptors in the microvascular to avoid the spleen passage, where they would be destroyed. Rosetting is where iRBC bind to uninfected red blood cells [44]. Sequestration and rosetting in *P. falciparum* are responsible for the pathogenicity of severe malaria. The extracellular region of *PfEMP1* contains several receptor binding domains. Duffy binding like (DBL) domains bind to intercellular adhesion molecule 1 (ICAM1) of the endothelial and the erythrocyte membrane protein, glycoprotein. The cysteine-rich interdomain regions (CIDR) of *PfEMP1* bind either to endothelial protein C receptor (EPCR) or CD36 [45]. The cysteine rich domain is also present on the extracellular region of SURFINs. However, from our result there was no significant difference of SURFIN$_{4.1}$ variants between clinical groups. Hence, function, endovascular receptor, and role of SURFINs in malaria infection and pathogenesis remains to be investigated.

Despite the sequence variation and variants diversity, predicted secondary protein structure of SURFIN$_{4.1}$ revealed five helix domains in the extracellular region conserved in all SURFIN$_{4.1}$ variants. The different of variant types were in the variable part which mostly strands and some coils. Comparing our secondary structure to Alphafold predicted tertiary structure of SURFIN$_{4.1}$ showed a globular protein where N-terminal is inside, and C-terminal is outside. In addition, phylogeny analysis of *surf$_{4.1}$* gene of this study and deposited sequences revealed SURFIN$_{4.1}$ could be grouped into four clades. Moreover, different SURFIN$_{4.1}$ variants were also found in the same clade. Thes suggests a conservation of extracellular domain among SURFIN$_{4.1}$ variants. Noteworthy, reference strains included in the analysis were isolated from different regions of Thailand such as North (Chiangmai), Northwest, near the Myanmar border (Tak and Kanchanaburi) and Northeast (Chaiyaphum and Saraburi) as well as different continents. Hence this also indicates SURFIN$_{4.1}$ variants were conserved across nationwide and continents.

Humoral immune response to malaria infection is a protective immunity. Protein expressed on the surface of infected red cells can trigger immune response are targeted for malaria vaccine development. However, a major obstacle in the development of malaria vaccine is its substantial level of antigenic diversity [46]. Our finding and others suggested extensive polymorphism of SURFIN$_{4.1}$ in both genetic and protein level. Nevertheless, we have identified several conserved and dimorphic epitopes as proposed vaccine candidates. Efficient antibody production composes of several processes, antigen recognition and presentation to T helper cells are one of the key steps. The proposed peptides were screened both for B cell epitope and helper T lymphocyte epitopes. Hence, they could induce immune responses to which could be effective against erythrocytic stage of multiple *P. falciparum* parasite strains in genetically diverse populations.

## Supporting information

**S1 Table. Deposited *Pfsurf$_{4.1}$* in GenBank and PlasmoDB database.**
(PDF)

**S2 Table. Alanine insertion causing different SURFIN$_{4.1}$ variants.**
(PDF)

**S3 Table. Difference of frequency between SURFIN$_{4.1}$, CNV and clinical outcomes.**
(PDF)

**S4 Table. Transmembrane topology location of SURFIN₄.₁.**
(PDF)

**S1 File. *Pfsurf4.1* nucleotide sequence alignment.**
(PDF)

**S2 File. SURFIN₄.₁ amino acid sequence alignment.**
(PDF)

**S3 File. *Pfsurf₄.₁* raw data.**
(XLSX)

## Acknowledgments

The authors wish to thank Faculty of Medicine, King Mongkut's Institute of Technology Ladkrabang for administrative assistance. We sincerely thank the malaria patients who participated in this study.

## Author Contributions

**Conceptualization:** Nitchakarn Noranate.

**Data curation:** Nitchakarn Noranate, Jariya Sripanomphong.

**Formal analysis:** Nitchakarn Noranate, Jariya Sripanomphong, Fingani Annie Mphande-Nyasulu, Suwanna Chaorattanakawee.

**Funding acquisition:** Nitchakarn Noranate.

**Investigation:** Nitchakarn Noranate, Jariya Sripanomphong.

**Methodology:** Nitchakarn Noranate.

**Project administration:** Nitchakarn Noranate.

**Resources:** Nitchakarn Noranate, Suwanna Chaorattanakawee.

**Software:** Nitchakarn Noranate.

**Supervision:** Nitchakarn Noranate.

**Validation:** Nitchakarn Noranate.

**Visualization:** Nitchakarn Noranate.

**Writing – original draft:** Nitchakarn Noranate.

**Writing – review & editing:** Nitchakarn Noranate, Fingani Annie Mphande- Nyasulu, Suwanna Chaorattanakawee.

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
