## [Decision Letter · Decision Letter 0]

16 Aug 2024

PONE-D-24-13672Plasmodium falciparum surf4.1 in clinical isolates: from genetic variation and variant diversity to in silico design immunopeptides for vaccine developmentPLOS ONE

Dear Dr. Noranate,

Thank you for submitting your manuscript to PLOS ONE. After careful consideration, we feel that it has merit but does not fully meet PLOS ONE’s publication criteria as it currently stands. Therefore, we invite you to submit a revised version of the manuscript that addresses the points raised during the review process.

We look forward to receiving your revised manuscript.

Kind regards,

Aditya K. Panda, Ph.D.

Academic Editor

PLOS ONE

Journal Requirements:

2. In the online submission form, you indicated that "The data will be available upon request"

Reviewers' comments:

Reviewer's Responses to Questions

**Comments to the Author**

1. Is the manuscript technically sound, and do the data support the conclusions?

Reviewer #1: Yes

Reviewer #2: Yes

2. Has the statistical analysis been performed appropriately and rigorously? 

Reviewer #1: Yes

Reviewer #2: Yes

3. Have the authors made all data underlying the findings in their manuscript fully available?

Reviewer #1: No

Reviewer #2: Yes

4. Is the manuscript presented in an intelligible fashion and written in standard English?

Reviewer #1: Yes

Reviewer #2: Yes

5. Review Comments to the Author

Reviewer #1: Comments to PONE-D-24-13672

Plasmodium falciparum surf4.1 in clinical isolates: from genetic variation and variant diversity to in silico design immunopeptides for vaccine development

The authors amplified and sequenced 76 clinical isolates of the Plasmodium falciparum surf4.1 gene obtained from patients in Thailand. The resulting sequences were analyzed for genetic diversity and epitope prediction. Upon reviewing the study, I found it potentially valuable for readers. Overall, the methodology, data analysis, and interpretation are adequate. However, several issues must be addressed before the manuscript is suitable for publication.

Abstract:

Line 25: "SURFINs family expressed…." should be "SURFINs protein family expressed…". Alternatively, please consider rewriting this sentence.

Line 28: "P. falciparum" should be italicized.

Lines 75-76: “Furthermore, we have identified four conserved and four dimorphic predicted epitopes as proposed vaccine candidates.” Please consider rewriting this sentence.

Introduction:

Line 50: "Erythrocytic" should be "erythrocytic".

Line 53: "P falciparum" should be "P. falciparum".

Lines 50-52 and 60-61: "Parasitized red blood cells……." are repetitive.

Lines 75-76: "High diversity of variable regions in exon 1 were reported in P. falciparum field isolates," please clarify the origin of these isolates.

Lines 80-81: "Still the causal of CNV has not been verified," please rewrite to improve clarity.

Materials and Methods:

Line 93: "P. falciparum infection was diagnosed by microscopic…." How did the authors exclude co-infection with other Plasmodium species, given that microscopic examination is not entirely reliable for species identification? Additionally, underlying diseases causing anemia might also play a role. Therefore, the clinical outcomes classified as complicated and uncomplicated malaria might be compromised.

Lines 118, 120, 128, 129, and possibly elsewhere throughout the manuscript: "Takarabio," please provide the country of the product manufacturer.

Line 139: "and" should not be italicized.

Line 155: ")" should be ".".

Line 175: "3bp" should be "3 bp".

Line 187: "Standard" should be "standard".

Lines 211, 218, and elsewhere: "outside domain," I suggest using "extracellular domain."

Results:

Lines 243-244: This sentence may not be necessary.

Table 1: "DNASP analysis nucleotide diversity," consider rewriting this table caption, e.g., "Nucleotide diversity indices of…."

Table 2: "Plasmodium falciparum" should be italicized, and "90bp, 3bp" should include a space, e.g., "90 bp, 3 bp".

Line 324: "substitution of 247 and 26 and synonymous," please correct.

Line 395: "Clade" should be "clade".

Line 398: "(" should be deleted.

Line 407: "conserved constructed among," please rewrite.

Discussion:

Line 465: “infected red cells” should be “infected red blood cells”

Reviewer #2: Dear Author

The research work was executed well and described the current frequency of genetic variation in SURFIN 4.1 in Thai clinical isolates. As this protein is an important vaccine candidate, such study is need of the moment. However, few clarifications on below facts will augment the quality of this research work;

1. Line 115 mentioned the PCR thermal profile in method section; it is not clear about the annealing time period, is it for 7-minutes @ 58 degree Celsius?

2. Line 140 mentioned about the concentration of dilutions used for PCR efficiency evaluation; Is the provided concentration of GENOMIC-DNA belongs to culture adapted falciparum or MR4 g-DNA is used here. If it is culture adapted falciparum, then please described how such concentration is achieved or if it is MR4 g-DNA, please acknowledge that.

3. It is not clear why n=46 was not sequenced for Exon-1 and performed for exon-2 only. The sequencing data of exon-1 could have provided comparable insight to the previous published sequence from Thai-Myanmar areas. Why this study is so focused on exon-2, when exon-1 is more variable, this can be discussed as a limitation of the study.

4. Line 523-524 and 530-531; cited Reference no 7 is not the correct one for the given statements/facts and can be replaced by reference no.6 the study related to Thai isolates.

5. Involvement of Thai isolates from other parts of the country for this type of study towards establishment of the SURFIN 4.1 as vaccine candidate for whole country may be discussed.

6. PLOS authors have the option to publish the peer review history of their article (what does this mean?). If published, this will include your full peer review and any attached files.

Reviewer #1: **Yes: **Morakot Kaewthamasorn

Reviewer #2: **Yes: **PRASHANT MALLICK

---

## [Author Response · Author response to Decision Letter 0]

14 Sep 2024

PONE-D-24-13672

Plasmodium falciparum surf4.1 in clinical isolates: from genetic variation and variant diversity to in silico design immunopeptides for vaccine development

PLOS ONE

Dear Dr. Noranate,

Thank you for submitting your manuscript to PLOS ONE. After careful consideration, we feel that it has merit but does not fully meet PLOS ONE’s publication criteria as it currently stands. Therefore, we invite you to submit a revised version of the manuscript that addresses the points raised during the review process.

We look forward to receiving your revised manuscript.

Kind regards,

Aditya K. Panda, Ph.D.

Academic Editor

PLOS ONE

Journal Requirements:

Answer: Following the guidelines, the authors have modified the manuscript according to PLOS ONE's style requirements i.e.

Formatting sample title authors affiliation

1. Delete postal codes and street address

2. Correcting format corresponding address

3. Remove Keywords, short running title, abbreviations

Formatting sample main body

4. Rename supplementary files

5. Place supplementary files after Reference section

6. Adding acknowledgements

2. In the online submission form, you indicated that "The data will be available upon request"

Answer: The authors will upload the data as supplementary information S3 file.

Answer: The corresponding author had provided the ORCID iD.

Answer: The author had written sentence” This study was approved (exempted) by the institutional review boards of King’s Mongkut Institute of Technology (EC-KMITL_63_054), Thailand. “

Answer: The author has added a separate caption for each figure in the manuscript.

Answer: The author has reviewed the reference list and made a change according to reviewer #2 suggested.

Reviewers' comments:

Reviewer's Responses to Questions

Comments to the Author

1. Is the manuscript technically sound, and do the data support the conclusions?

Reviewer #1: Yes

Reviewer #2: Yes

2. Has the statistical analysis been performed appropriately and rigorously?

Reviewer #1: Yes

Reviewer #2: Yes

3. Have the authors made all data underlying the findings in their manuscript fully available?

Reviewer #1: No

Reviewer #2: Yes

4. Is the manuscript presented in an intelligible fashion and written in standard English?

Reviewer #1: Yes

Reviewer #2: Yes

5. Review Comments to the Author

Reviewer #1: Comments to PONE-D-24-13672

Plasmodium falciparum surf4.1 in clinical isolates: from genetic variation and variant diversity to in silico design immunopeptides for vaccine development

The authors amplified and sequenced 76 clinical isolates of the Plasmodium falciparum surf4.1 gene obtained from patients in Thailand. The resulting sequences were analyzed for genetic diversity and epitope prediction. Upon reviewing the study, I found it potentially valuable for readers. Overall, the methodology, data analysis, and interpretation are adequate. However, several issues must be addressed before the manuscript is suitable for publication.

Abstract:

Line 25: "SURFINs family expressed…." should be "SURFINs protein family expressed…". Alternatively, please consider rewriting this sentence.

Answer: Corrected accordingly.

Line 28: "P. falciparum" should be italicized.

Answer: Corrected accordingly.

Lines 75-76: “Furthermore, we have identified four conserved and four dimorphic predicted epitopes as proposed vaccine candidates.” Please consider rewriting this sentence.

Answer: The sentense is in lines 45-46 and corrected accordingly.

Introduction:

Line 50: "Erythrocytic" should be "erythrocytic".

Answer: Corrected accordingly.

Line 53: "P falciparum" should be "P. falciparum".

Answer: Corrected accordingly.

Lines 50-52 and 60-61: "Parasitized red blood cells……." are repetitive.

Answer: Corrected accordingly.

Lines 75-76: "High diversity of variable regions in exon 1 were reported in P. falciparum field isolates," please clarify the origin of these isolates.

Answer: Corrected accordingly.

Lines 80-81: "Still the causal of CNV has not been verified," please rewrite to improve clarity.

Answer: corrected and rewritten.

Materials and Methods:

Line 93: "P. falciparum infection was diagnosed by microscopic…." How did the authors exclude co-infection with other Plasmodium species, given that microscopic examination is not entirely reliable for species identification? Additionally, underlying diseases causing anemia might also play a role. Therefore, the clinical outcomes classified as complicated and uncomplicated malaria might be compromised.

Answer: All samples are mono P. falciparum infection, based on microscopic blood examination. As the reviewer mentioned, we cannot exclude the co-infection with other species. However, the main objective of this study is to analyze the genetic diversity and identify immunogenic epitope for the vaccine candidate. We think the coinfection with other malaria species may not confound our main findings. Moreover, the role of mixed spp. infection on the clinical outcome remained unclear. Nonetheless, the underlining conditions of the patients may confound the clinical outcomes, this is a limitation of this study. 

Lines 118, 120, 128, 129, and possibly elsewhere throughout the manuscript: "Takarabio," please provide the country of the product manufacturer.

Answer: corrected-the author have provided the country of product manufacturer.

Line 139: "and" should not be italicized.

Answer: Corrected accordingly.

Line 155: ")" should be ".".

Answer: Corrected accordingly.

Line 175: "3bp" should be "3 bp".

Answer: Corrected accordingly.

Line 187: "Standard" should be "standard".

Answer: Corrected accordingly.

Lines 211, 218, and elsewhere: "outside domain," I suggest using "extracellular domain."

Answer: Corrected accordingly.

Results:

Lines 243-244: This sentence may not be necessary.

Table 1: "DNASP analysis nucleotide diversity," consider rewriting this table caption, e.g., "Nucleotide diversity indices of…."

Answer: Corrected accordingly.

Table 2: "Plasmodium falciparum" should be italicized, and "90bp, 3bp" should include a space, e.g., "90 bp, 3 bp".

Answer: Corrected accordingly.

Line 324: "substitution of 247 and 26 and synonymous," please correct.

Answer: Corrected accordingly.

Line 395: "Clade" should be "clade".

Answer: Corrected accordingly.

Line 398: "(" should be deleted.

Answer: corrected- the author have deleted the bracket.

Line 407: "conserved constructed among," please rewrite.

Answer: corrected-the author have re-written the sentence.

Discussion:

Line 465: “infected red cells” should be “infected red blood cells”

Answer: Corrected accordingly.

Reviewer #2: Dear Author

The research work was executed well and described the current frequency of genetic variation in SURFIN 4.1 in Thai clinical isolates. As this protein is an important vaccine candidate, such study is need of the moment. However, few clarifications on below facts will augment the quality of this research work;

1. Line 115 mentioned the PCR thermal profile in method section; it is not clear about the annealing time period, is it for 7-minutes @ 58 degree Celsius?

Answer: Yes. The PCR condition is following the polymerase enzyme manufacturer’s manual instruction.

2. Line 140 mentioned about the concentration of dilutions used for PCR efficiency evaluation; Is the provided concentration of GENOMIC-DNA belongs to culture adapted falciparum or MR4 g-DNA is used here. If it is culture adapted falciparum, then please described how such concentration is achieved or if it is MR4 g-DNA, please acknowledge that.

Answer: Corrected by adding sentences below in the material and method section.

Culture adapted P. falciparum 3D7 with 2% parasitemia were used to prepare genomic DNA using QIAamp DNA Blood mini kit following the manufacture’s manual instruction (Hilden, Germany). DNA concentration was measured by Implean NanoPhotometer N60/N50 (Munich, Germany).

3. It is not clear why n=46 was not sequenced for Exon-1 and performed for exon-2 only. The sequencing data of exon-1 could have provided comparable insight to the previous published sequence from Thai-Myanmar areas. Why this study is so focused on exon-2, when exon-1 is more variable, this can be discussed as a limitation of the study.

Answer: The amplified PCR products of full length surf4.1 (6757 bp) of 76 samples were subjected for sequencing. We were able to obtain complete full-length (exon1-intron-exon2) in 30 samples and partial sequences in 46 samples. Of those partial sequences, the obtained sequencing results showed a stretch of ambiguous code (NNNNNN) in variable region of exon1 and complete nucleotide sequence of intron and exon 2 (partial exon1-intron-exon2. The sequence of full-length nucleotide sequences (n=30) were used to analyse genetic variation and provided comparable insight to the previous published sequences from Thailand and worldwide. The exon2 provide insight information to identify type of SURFIN4.1 variants, truncated or contain different number of tryptophan rich domain. This will affect location of SURFIN4.1 protein expression. The TM variants will only expressed on the merozoite surface while WDs variants will expressed on both merozoite and infected red blood cell surface. 

To clarify this point , we’ve re-written the sentence to “From 76 clinical isolates, 30 complete full-length and 46 partial nucleotide sequences of surf4.1 gene were obtained in this study.” in results under section: The organization of full-length surf4.1 gene (exon1-intron-exon2) and genetic variation of coding sequence.

4. Line 523-524 and 530-531; cited Reference no 7 is not the correct one for the given statements/facts and can be replaced by reference no.6 the study related to Thai isolates.

Answer: Corrected- reference 7 has been replaced by reference 6

5. Involvement of Thai isolates from other parts of the country for this type of study towards establishment of the SURF

---

## [Decision Letter · Decision Letter 1]

2 Oct 2024

Plasmodium falciparum surf4.1 in clinical isolates: from genetic variation and variant diversity to in silico design immunopeptides for vaccine development

PONE-D-24-13672R1

Dear Dr. Noranate,

We’re pleased to inform you that your manuscript has been judged scientifically suitable for publication and will be formally accepted for publication once it meets all outstanding technical requirements.

Kind regards,

Aditya K. Panda, Ph.D.

Academic Editor

PLOS ONE

Reviewers' comments:

Reviewer's Responses to Questions

**Comments to the Author**

1. If the authors have adequately addressed your comments raised in a previous round of review and you feel that this manuscript is now acceptable for publication, you may indicate that here to bypass the “Comments to the Author” section, enter your conflict of interest statement in the “Confidential to Editor” section, and submit your "Accept" recommendation.

Reviewer #1: All comments have been addressed

Reviewer #2: All comments have been addressed

2. Is the manuscript technically sound, and do the data support the conclusions?

Reviewer #1: Yes

Reviewer #2: Yes

3. Has the statistical analysis been performed appropriately and rigorously? 

Reviewer #1: Yes

Reviewer #2: Yes

4. Have the authors made all data underlying the findings in their manuscript fully available?

Reviewer #1: Yes

Reviewer #2: Yes

5. Is the manuscript presented in an intelligible fashion and written in standard English?

Reviewer #1: Yes

Reviewer #2: Yes

6. Review Comments to the Author

Reviewer #1: I thank the authors for adequately addressing all reviewers' comments, and I recommend the manuscript for publication.

Reviewer #2: (No Response)

7. PLOS authors have the option to publish the peer review history of their article (what does this mean?). If published, this will include your full peer review and any attached files.

Reviewer #1: **Yes: **Morakot Kaewthamasorn

Reviewer #2: **Yes: **PRASHANT MALLICK

---

## [Editor Report · Acceptance letter]

9 Oct 2024

PONE-D-24-13672R1 

PLOS ONE

Dear Dr. Noranate, 

I'm pleased to inform you that your manuscript has been deemed suitable for publication in PLOS ONE. Congratulations! Your manuscript is now being handed over to our production team.

Kind regards, 

on behalf of

Dr. Aditya K. Panda 

Academic Editor

PLOS ONE